# Cryo-EM structures of the SARS-CoV-2 endoribonuclease Nsp15 reveal insight into nuclease specificity and dynamics

Monica C. Pillon [1,7,8✉], Meredith N. Frazier[1,8], Lucas B. Dillard[2,8], Jason G. Williams[3], Seda Kocaman[1], Juno M. Krahn[2], Lalith Perera [2], Cassandra K. Hayne [1], Jacob Gordon [1,4,5,6], Zachary D. Stewart[1], Mack Sobhany[1], Leesa J. Deterding[3], Allen L. Hsu [2], Venkata P. Dandey[2], Mario J. Borgnia [2] & Robin E. Stanley [1✉]

Nsp15, a uridine specific endoribonuclease conserved across coronaviruses, processes viral RNA to evade detection by host defense systems. Crystal structures of Nsp15 from different coronaviruses have shown a common hexameric assembly, yet how the enzyme recognizes and processes RNA remains poorly understood. Here we report a series of cryo-EM reconstructions of SARS-CoV-2 Nsp15, in both apo and UTP-bound states. The cryo-EM reconstructions, combined with biochemistry, mass spectrometry, and molecular dynamics, expose molecular details of how critical active site residues recognize uridine and facilitate catalysis of the phosphodiester bond. Mass spectrometry revealed the accumulation of cyclic phosphate cleavage products, while analysis of the apo and UTP-bound datasets revealed conformational dynamics not observed by crystal structures that are likely important to facilitate substrate recognition and regulate nuclease activity. Collectively, these findings advance understanding of how Nsp15 processes viral RNA and provide a structural framework for the development of new therapeutics.

[1] Signal Transduction Laboratory, National Institute of Environmental Health Sciences, National Institutes of Health, Department of Health and Human Services, 111 T. W. Alexander Drive, Research Triangle Park, NC 27709, USA. [2] Genome Integrity and Structural Biology Laboratory, National Institute of Environmental Health Sciences, National Institutes of Health, Department of Health and Human Services, 111 T. W. Alexander Drive, Research Triangle Park, NC 27709, USA. [3] Epigenetics and Stem Cell Biology Laboratory, National Institute of Environmental Health Sciences, National Institutes of Health, Department of Health and Human Services, 111 T. W. Alexander Drive, Research Triangle Park, NC 27709, USA. [4] Cambridge Institute for Medical Research, Cambridge, UK. [5] Department of Haematology, University of Cambridge, Cambridge, UK. [6] Wellcome Trust-Medical Research Council Stem Cell Institute, University of Cambridge, Cambridge, UK. [7] Present address: Department of Biochemistry and Molecular Biology, Baylor College of Medicine, 1 Baylor Plaza, Houston, Texas 77030, USA. [8] These authors contributed equally: Monica C. Pillon, Meredith N. Frazier, Lucas B. Dillard. ✉email: monica.pillon@nih.gov; robin.stanley@nih.gov

Severe acute respiratory syndrome coronavirus 2 (SARS-CoV-2) is the virus at the center of the unprecedented Covid-19 global health pandemic. All coronaviruses have large single-stranded, positive-sense RNA genomes that harbor two open reading frames that are translated upon entry into the host (Supplementary Fig. 1). The open reading frames in SARS-CoV-2 are translated by host ribosomes into two long poly-proteins that are processed by viral proteases into distinct non-structural proteins (designated by the Nsp acronym). These non-structural proteins serve a variety of roles in virus repli-cation, virus assembly, and evasion of host viral sensors, but the precise function of many of these proteins are poorly understood[1]. Although some inhibitors show promise as anti-viral drugs directed specifically against SARS-CoV-2 non-structural proteins, there is a pressing need to understand the structure and function of the non-structural proteins to aid in the development of new and effective antiviral therapies for Covid-19[2].

Nsp15-like endoribonucleases are a characteristic of all cor-onavirus family members[3]. Biochemical experiments with recombinant Nsp15 have established that it preferentially cleaves RNA substrates 3′ of uridines, and therefore Nsp15 is commonly called endoU alluding to its cleavage specificity[4–7]. Conservation of Nsp15 across Coronaviridae suggests that its endonuclease function is critical for their viral life cycle, however, the specific role of Nsp15 in viral propagation is still unclear. Nsp15 was initially thought to play an essential proofreading role in viral replication until it was shown that the coronavirus, mouse hepatitis virus (MHV), can replicate with a catalytically deficient variant of Nsp15 in cell culture[3,6]. More recent work suggests that rather than functioning in viral RNA synthesis, Nsp15 nuclease activity is important to evade activation of host immune responses[8–11]. Loss of Nsp15 nuclease activity in porcine epi-demic diarrhea coronavirus (PEDV) leads to the activation of interferon responses and reduced viral shedding and mortality in infected piglets[9]. Similar results have also been observed in mice infected with Nsp15 nuclease deficient strains of MHV, which display attenuated viral replication[10–12]. Recent analysis of viral RNA from MHV-infected cells harboring a catalytically deficient Nsp15 revealed an accumulation of 12–17 polyuridine tracts at the 5′-end of the negative-strand viral RNA intermediates. Con-sidering that polyuridine negative-strand RNA elicits an interferon-mediated response, this suggests a role for Nsp15 in regulating the length of polyuridines found at the 5′-end of negative-strand viral RNA to evade activation of the host innate immune response (Supplementary Fig. 1)[8]. High-throughput sequencing from MHV-infected macrophages also recently identified additional Nsp15 cleavage sites within the viral positive-strand RNA (Supplementary Fig. 1)[13]. Collectively, these studies suggest the existence of multiple Nsp15 cleavage targets that are important to regulate the accumulation of viral RNA and prevent activation of RNA-activated antiviral responses. Nsp15 may therefore be a suitable target for an antiviral small molecule inhibitor or the development of an attenuated live virus vaccine[8,9].

Crystal structures of Nsp15 have been reported for a number of Coronaviridae, but there are no high-resolution cryo-EM struc-tures of Nsp15. The crystal structures of Nsp15 have revealed a common hexameric assembly made up of a dimer of Nsp15 trimers[4,5,14–16]. Each Nsp15 protomer is composed of three domains including an N-terminal domain (ND) that is important for oligomerization, a variable middle domain (MD), and a C-terminal endonuclease (endoU) domain that shares homology with other endoU enzymes (Fig. 1a)[17]. Nsp15 is only active as a hexamer, but the molecular requirement for Nsp15 oligomer-ization is unknown. A crystal structure of Nsp15 from SARS-CoV-1 lacking the first 28 residues of the ND is monomeric and reveals a misfolded endoU active site suggesting that Nsp15 may rely on oligomerization as an allosteric activation switch[15]. How Nsp15 specifically cleaves the phosphodiester bond following uridines is also poorly understood. The Nsp15 active site contains two well-conserved histidine residues that are important for catalysis and reminiscent of the well-characterized RNase A active site[4,14,16,18]. RNase A catalyzes a two-step reaction that first generates a 2′3′-cyclic phosphate (2′3′-cP) that is then hydrolyzed to form a 3′-phosphate (3′-P)[19]. It is currently unclear if Nsp15 facilitates the second hydrolysis step like RNase A[20,21]. To gain insight into the structure and catalytic mechanism of Nsp15, we solved a series of cryo-electron microscopy (cryo-EM)recon-structions of Nsp15 from SARS-CoV-2. The cryo-EM-derived atomic model of Nsp15 bound to UTP identified residues important for uridine recognition. Reconstructions of Nsp15 in the absence of ligand and molecular dynamic simulations uncovered the conformational malleability of the endoU domain. Furthermore, biochemistry and mass spectrometry revealed molecular details into how the Nsp15 hexamer processes RNA.

## Results

**Cryo-EM reconstruction of SARS-CoV-2 Nsp15**. We char-acterized SARS-CoV-2 Nsp15 bound to a UTP nucleotide by single particle cryo-EM. Recombinant wild-type (wt) Nsp15 was purified as a stable hexamer using a bacterial expression system (Supplementary Fig. 2). Over 1000 micrographs were collected from grids prepared with recombinant wt-Nsp15 and an excess of UTP. Following 3D classification, the particles converged into a single prominent class with a resolution of 3.38 Å (Supplementary Fig. 3). The cryo-EM reconstruction of wt-Nsp15 bound to UTP revealed a hexameric assembly containing six protomers of Nsp15 (designated as P1–P6) (Fig. 1b) with D3 symmetry. Classification and refinement were performed with and without imposing D3 symmetry, however, no asymmetric conformational states were observed. A combination of rigid-body and real-space refinement was used to fit the crystal structure of SARS-CoV-2 Nsp15 (PDB ID: 6WLC) into the cryo-EM reconstruction (Table 1).

For each Nsp15 molecule, all three domains of Nsp15 were clearly visible in the reconstruction as well as the uracil base and ribose sugar of the bound UTP molecule (Supplementary Fig. 3). Despite adding excess manganese to the storage buffer, we did not observe discrete density for a metal ion suggesting that the stabilizing effect of manganese may be due to non-specific interactions along the highly charged patches of Nsp15[4,22]. Overall, the cryo-EM structure of Nsp15 is very similar to recently published crystal structures of SARS-Cov-2 Nsp15[14]. The barrel-shaped Nsp15 particle has dimensions of $100 \times 110$ Å with a narrow, negatively charged channel that runs through the middle (Fig. 1b). The small ND domain, composed of 64 amino acid residues, mediates oligomerization of the complex by forming two head-to-head stacked trimers. Following the ND is the MD, composed of a mixture of beta-strands and two small alpha-helices (Fig. 1c). Following the MD is the well-conserved endoU domain containing the nuclease active site (Fig. 1c). The unique arrangement of the six individual Nsp15 protomers positions the endoU domains on opposite ends of the hexamer (Supplementary Fig. 4).

**UTP-bound structure reveals the basis for uridine specificity**. The Nsp15 active site contains several residues that are well conserved amongst Nsp15 homologs and important for nuclease activity and specificity (Fig. 2a and Supplementary Fig. 5). The active site within each individual Nsp15 protomer lies near the

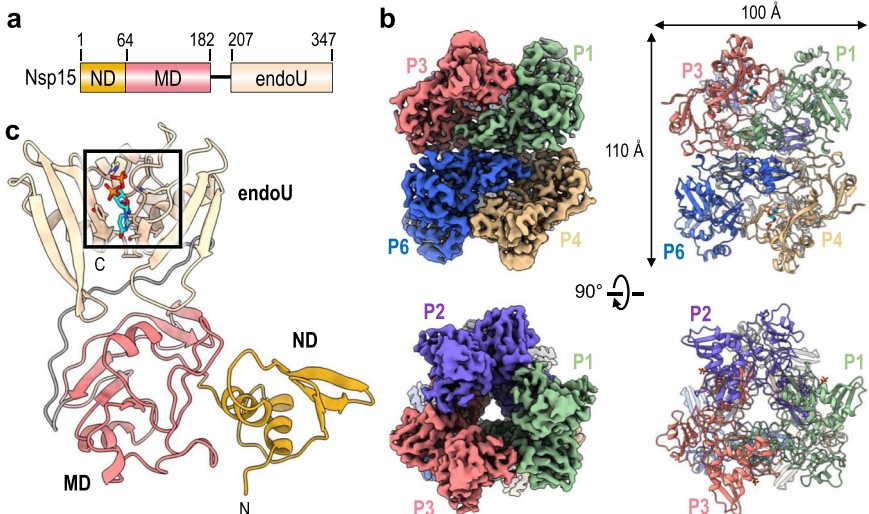

**Fig. 1 Architecture of hexameric SARS-CoV-2 Nsp15. a** Domain organization of Nsp15. The numbering corresponds to the amino acid residues at the domain boundaries. The Nsp15 N-terminal domain (ND) is shown in orange, the middle domain (MD) is shown in red, and the poly-U-specific endonuclease domain (endoU) is shown in beige. **b** Orthogonal views of the cryo-EM map reconstruction of UTP-bound Nsp15 (left) and its corresponding model shown as a cartoon (right). Each Nsp15 protomer is colored as green (P1), purple (P2), coral (P3), tan (P4), gray (P5), and blue (P6). **c** Model of Nsp15 protomer shown as a cartoon and colored as seen in (**a**). The position of the Nsp15 endoU active site is highlighted with a black box and the modeled nucleotide is shown as sticks (cyan). N and C mark the N- and C-termini, respectively.

interface with a neighboring endoU domain (Fig. 2b). This close positioning of the active site to the neighboring protomer hints at a possible mechanism of allosteric communication between the Nsp15 protomers. The active site residues within the endoU domain have well-defined side-chain density in the cryo-EM reconstruction along with additional density for a ligand (Fig. 2b, Supplementary Fig. 4, and Supplementary Movie 1). While we used an excess of UTP when vitrifying Nsp15, we modeled uridine 5′-monophosphate (5′-UMP) into the active site because there was no observable density to account for the β- and γ-phosphates (Fig. 2c and Supplementary Fig. 3). We observed additional ambiguous density next to 5′-UMP, which was modeled as a phosphate ion (Fig. 3c; model 3′-PO₄) (Supplementary Fig. 3). Our modeled phosphate is in good agreement with a recent, deposited structure of SARS-CoV-2 Nsp15 with uridine-2′,3′,-vanadate, which mimics a cyclic intermediate (Supplementary Fig. 6). The ribose sugar of uridine forms van der Waals interactions with Y343, suggesting that this residue is critical for orienting the ribose within the active site for cleavage (Fig. 2c; top). Mutation of the equivalent tyrosine to alanine in SARS-CoV-1 and MERS (Middle East Respiratory Syndrome) Nsp15 leads to an almost complete loss in nuclease activity, underscoring the importance of this tyrosine residue[4,5].

Calculation of the electrostatic surface potential revealed that aside from the small electropositive pocket surrounding the active site, the surface of Nsp15 is negatively charged (Supplementary Fig. 4c). Using the position of the UMP as a guide we modeled a small dinucleotide into the active site. Beyond the active site pocket, there are no obvious RNA binding surfaces along the Nsp15 hexamer. Nsp15 may be reliant on additional viral factors such as Nsp7 and Nsp8 to facilitate viral RNA recruitment[4,23].

The cryo-EM structure revealed the significance of S294 for uridine specificity. Molecular docking of 5′-UMP into the Nsp15 active site previously suggested that this serine residue may be the critical determinant for base specificity[5]. The O2 and N3 positions of the uracil base are within hydrogen bonding distance of the S294 backbone N and side-chain OG atoms, respectively (Fig. 2c; bottom). Nsp15 N278 is also within hydrogen bonding distance of S294 and likely contributes to proper positioning of

S294 within the active site. Across Nsp15 homologs, the equivalent residue to S294 is either a serine or threonine (Fig. 2a). Previous work with SARS-CoV-1 Nsp15 demonstrated that mutation of this serine to a threonine does not affect activity. Yet, mutation to an alanine reduces cleavage activity by more than 5-fold and abrogates uridine specificity such that a single rU or rC containing deoxyribonucleotides are cleaved at comparable rates[5]. While this manuscript was in preparation, a series of additional Nsp15 crystal structures with different ligands were deposited in the Protein Data Bank (PDB ID 6WLC)[24]. One of these structures includes a 5′-UMP in the active site. This crystal structure is in excellent agreement (RMSD of 0.35 Å across all residues in chain A) with our cryo-EM reconstruction and supports the same conclusions drawn about the role of S294 in base specificity. The crystal structure also reveals L346 is within hydrogen bonding distance with O4 of the uracil base and likely contributes to active site alignment along with uridine specificity. Interestingly, L346 is not ordered in our cryo-EM reconstruction suggesting substrate binding may promote additional ordering of the active site. Thus, our cryo-EM reconstruction, recent crystal structures, and previous biochemical work support a prominent role for S294 in uracil base discrimination within the Nsp15 active site.

**Two active site histidine residues are required for nuclease activity**. The active site of Nsp15 contains three residues that are required for catalysis including two histidine residues and a lysine residue, which is analogous to the active site of RNase A. Within the RNase A active site, one histidine functions as a general base to activate the 2′OH while the other histidine functions as a general acid to donate a proton to the leaving 5′OH[19]. During the second hydrolysis step, the roles of the two active site histidine residues are reversed. The three Nsp15 catalytic residues H235, H250, and K290 are clustered around the ribose sugar of UMP (Fig. 2c; top). H250 and K290 are both within hydrogen bonding distance of the 2′OH of the ribose. The active site arrangement suggests that similar to RNase A, H250 functions as a general base to activate the 2′OH for nucleophilic attack, while K290 stabilizes

**Table 1 Cryo-EM data collection, refinement, and validation statistics.**

| EMDB PDB ID | UTP-bound wt-Nsp15 22610 7K0R | wt-Nsp15 22611 | Nsp15 H235A dataset i 22613 | Nsp15 H235A dataset ii 22612 |
|---|---|---|---|---|
| **Data collection and processing** | | | | |
| Magnification | 36,000 | 36,000 | 45,000 | 45,000 |
| Voltage (kV) | 200 | 200 | 200 | 200 |
| Electron exposure ($e^-/Å^2$) | ~54 | ~54 | ~54 | ~54 |
| Defocus range (μm) | −1.0 to −2.5 | −1.0 to −2.5 | −1.0 to −2.5 | −1.0 to −2.5 |
| Pixel size (Å) | 1.187 | 1.187 | 0.932 | 0.932 |
| Scaled pixel size (Å) | 1.145 | 1.145 | 0.8991 | 0.8991 |
| Symmetry imposed | D3 | D3 | D3 | D3 |
| Initial particle images (no.) | 383,271 | 672,879 | 382,348 | 641,748 |
| Final particle images (no.) | 65,393 | 167,106 | 59,703 | 433,403 |
| Map resolution (Å) (FSC = 0.143) | 3.38 | 3.02 | 3.37 | 2.94 |
| Map resolution (Å) (FSC = 0.5) | 3.62 | 3.34 | 3.59 | 3.04 |
| Map resolution range (Å) | 3.3 to 6.2 | 2.6 to 4.9 | 2.8 to 6.0 | 2.3 to 5.8 |
| **Refinement** | | | | |
| *Model composition* | | | | |
| Nonhydrogen atoms | 16,038 | | | |
| Protein residues | 2070 | | | |
| Ligands | 12 | | | |
| *B factors ($Å^2$)* | | | | |
| Protein | 99.24 | | | |
| Ligand | 131.77 | | | |
| *Map-model CC* | | | | |
| CC (mask) | 0.8335 | | | |
| CC (volume) | 0.8287 | | | |
| CC (peaks) | 0.7920 | | | |
| CC (box) | 0.8794 | | | |
| *R.M.S. deviations* | | | | |
| Bond lengths (Å) | 0.009 | | | |
| Bond angles (°) | 0.767 | | | |
| *Validation* | | | | |
| Molprobity score | 1.23 | | | |
| Clashscore | 3.56 | | | |
| Poor rotamers (%) | 1.05 | | | |
| *Ramachandran plot* | | | | |
| Favored (%) | 97.67 | | | |
| Allowed (%) | 2.33 | | | |
| Disallowed (%) | 0 | | | |

the deprotonation of O2′. In contrast, H235 is set back further from the ribose and is within hydrogen bonding distance of the phosphate ion that we modeled into the active site (Fig. 2c; top). This modeled phosphate is adjacent to the 3′OH of the ribose and mimics the position of the phosphate group to be attacked by the 2′OH. Based on the active site arrangement, H235 is properly aligned to play the role of the general acid during the transesterification reaction.

To verify that our recombinant Nsp15 retained nuclease activity and to confirm the significance of the SARS-CoV-2 active site histidine residues, we adapted a FRET-based assay previously used to measure the activity of Nsp15 from MERS coronavirus[4]. We used a short 6-mer oligonucleotide (5′-AAAUAA) that is cleaved by Nsp15 3′ to the single rU. The substrate contains 5′-fluorescein (FI) and 3′-TAMRA labels. The FI fluorescence is quenched by the TAMRA label in the uncleaved substrate, and we can monitor RNA cleavage by measuring the increase in FI fluorescence as the TAMRA label is released. We made two single mutants of Nsp15 in which we individually mutated the active site histidine residues to alanine (H235A or H250A). Both mutants purified as stable hexamers, confirming that these mutations do not disrupt the oligomerization of Nsp15 (Supplementary Fig. 2). We measured RNA cleavage over a

one-hour time course with wt-Nsp15 and the H235A and H250A Nsp15 variants. Wt-Nsp15 displayed robust RNA cleavage that increases as a function of time, but we could not detect any RNA cleavage with the two histidine mutants (Fig. 2d). These results confirm that recombinant Nsp15 is active and H235 and H250 are required for RNA cleavage.

**Nsp15 predominantly catalyzes 2′-*O*-transesterification**. After confirming that our recombinant Nsp15 was active, we used mass spectrometry to determine the identity of the Nsp15 RNA cleavage products. Mass spectrometry can identify the site of RNA cleavage and distinguish the mass difference of 18.01 Da for the two possible 3′-end products (the 2′3′-cP intermediate versus the 3′-P hydrolyzed intermediate). First, we calculated the theoretical masses of all potential cleavage products from our 6-mer FRET RNA substrate (Supplementary Table 1). We analyzed the 6-mer FRET RNA substrate using liquid chromatography electrospray ionization mass spectrometry (LC-ESI-MS) in the absence of Nsp15 and observed a MS spectra peak of 3294.74 Da, corresponding to the theoretical molecular mass of the uncleaved RNA substrate (Supplementary Fig. 7a and Supplementary Table 1). Next, we analyzed the RNA products following a 30 min

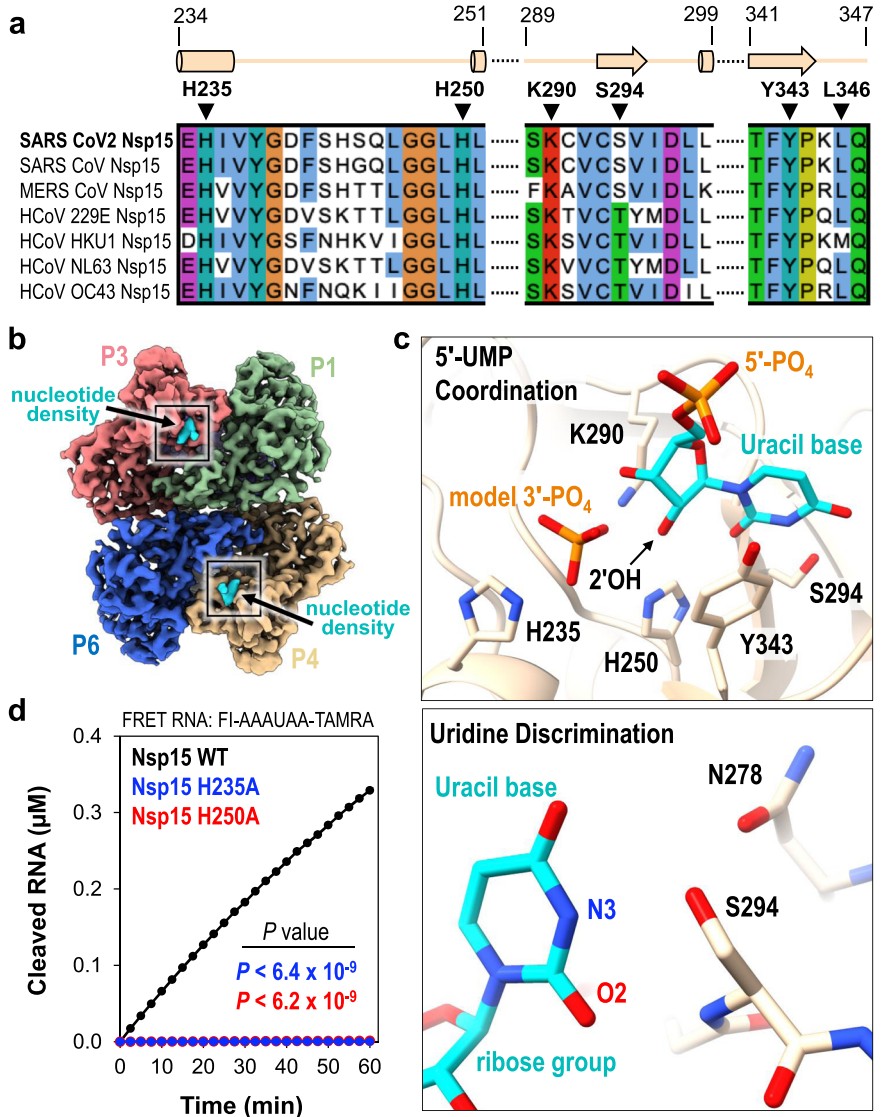

**Fig. 2 5′-UMP coordination by the Nsp15 endoU active site. a** Amino acid sequence alignment of endoU active site residues from Nsp15 homologs. Secondary structure motifs observed in the nucleotide-bound Nsp15 cryo-EM structure are shown above with their corresponding amino acid residue boundaries. **b** Nucleotide-bound Nsp15 cryo-EM map reconstruction with protomers colored as seen in Fig. 1b. Excess UTP was added to the sample resulting in additional density within all six endoU active sites. The nucleotide density is colored in cyan and the black box demarcates the endoU active site. **c** Nsp15 coordination of 5′-UMP ligand. Due to the poor density of the UTP β- and γ-phosphates, 5′-UMP was modeled into the active site. Cartoon model of the 5′-UMP-bound Nsp15 endoU active site where 5′-UMP (cyan) and individual residues H235, H250, K290, S294, and Y343 are shown as sticks (top). Model of uracil base discrimination shown as sticks (bottom). **d** RNA cleavage activity of Nsp15 variants (2.5 nM) incubated with FRET RNA substrate (0.8 μM) over time. RNA cleavage was quantified from three technical replicates. The mean and standard deviation are plotted and P values of wt-Nsp15 compared to H235A (blue; $P < 6.4 \times 10^{-9}$) and H250A (red; $P < 6.2 \times 10^{-9}$) are reported from two-tailed Student's t tests.

enzymatic reaction with Nsp15. Similar to other Nsp15 homologs, we observed products resulting from a single cleavage site 3′ to the uridine (5′-AAAU^AA), confirming Nsp15's strict specificity for uridine. We observed a prominent peak in the extracted ion chromatogram with a calculated mass of ~1463 Da, which was absent in our negative control (Fig. 3a). The experimental mass of this RNA product corresponds well with the theoretical mass of 5′-HO-AA-TAMRA (Supplementary Table 1). The MS spectrum of the 5′-HO-AA-TAMRA product has a mass of 1464.43 Da, and in conjunction with MSMS data confirms its identity (Fig. 3b and Supplementary Fig. 7b). Furthermore, we could not detect any other 5′ cleavage products, confirming Nsp15 exclusively cuts the RNA substrate 3′ of the uridine. A second prominent peak was also detected in the extracted ion chromatogram that was absent in our negative control and

corresponds to a 3′ RNA product (Fig. 3c). The MS spectrum of the 3′ product revealed a doubly charged ion at m/z 914.14 and, hence, a molecular mass of 1830.30 Da corresponding to 5′-Fl-AAAU-2′3′-cP and another doubly charged ion at m/z 923.14 and, hence, a molecular mass of 1848.31 Da corresponding to 5′-Fl-AAAU-3′-P (Fig. 3d, Supplementary Fig. 7c and Supplementary Table 1). To determine which 3′ RNA product is more prominent, we normalized and compared the abundance of the 2′3′-cP and 3′-P products assuming similar ionization efficiencies of the 2′3′-cP and 3′-P products (Fig. 3e). We observed that the 2′3′-cP makes up 80% of the total 3′-product confirming the 2′3′-cP is the major cleavage product and suggesting the rate of hydrolysis is slow (Fig. 3f).

We compared the active sites of Nsp15 and RNase A to determine why Nsp15 does not promote the hydrolysis step as

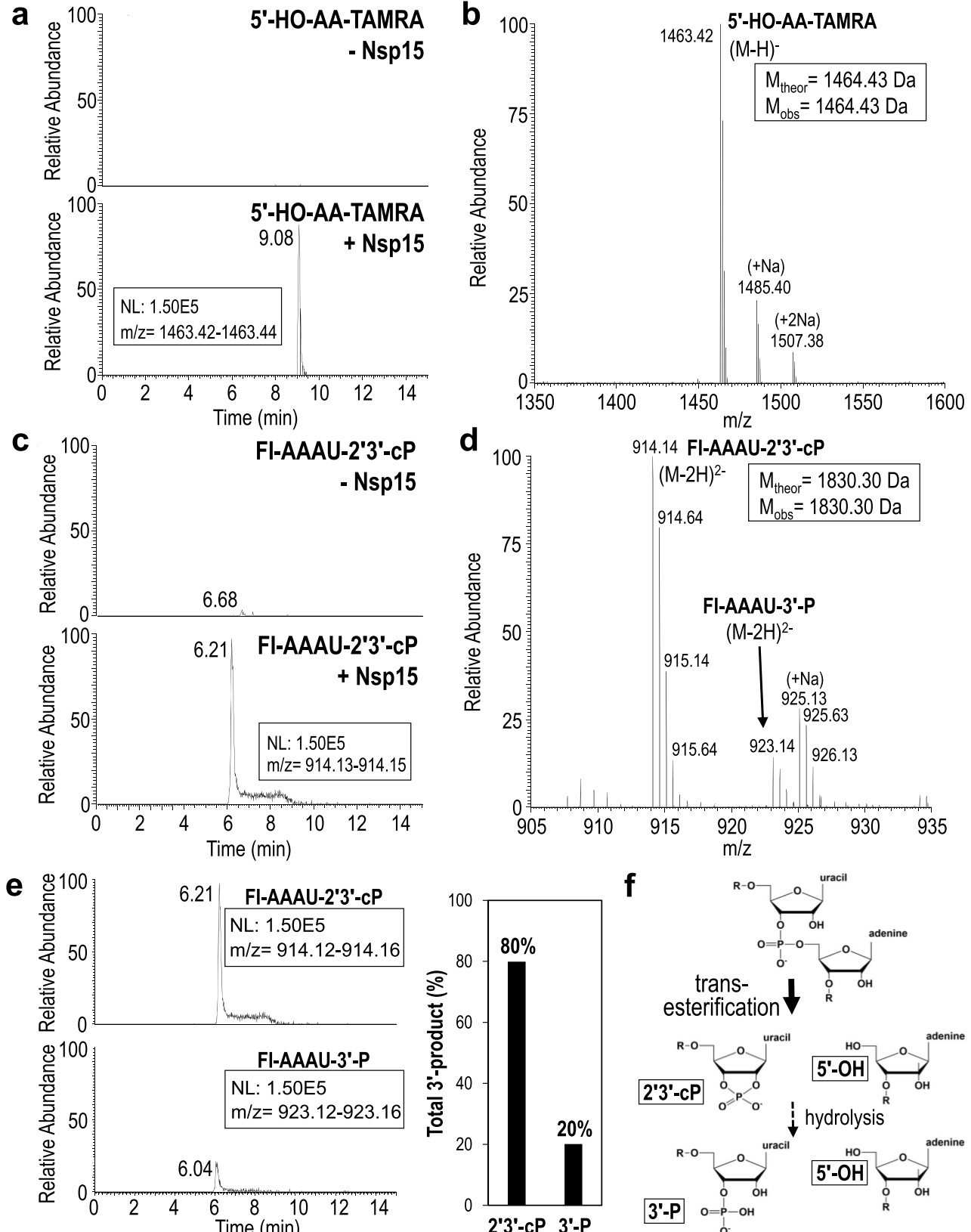

efficiently as RNase A. We superimposed the active sites of Nsp15 and RNase A using a crystal structure of RNase A solved in the presence of 3′-UMP[25]. H250 and K290 of Nsp15 superimpose well with the equivalent H12 and K41 residues from RNase A (Fig. 4a, b). In contrast, H235 from Nsp15 and H119 from RNase A do not superimpose (Fig. 4b). H119 of RNase A is near the ribose group and is fixed into position by a network of hydrogen bonds and tightly coupled to a carboxyl (D121) to promote catalytic action. Conversely, H235 of Nsp15 is ~ 8 Å away from the ribose group and therefore it is not positioned to promote

**Fig. 3 Extracted ion chromatograms and MS spectra of Nsp15 RNA cleavage products. a** Extracted ion chromatograms of the 3'-TAMRA labeled AA-TAMRA RNA cleavage product. 5'-HO-AA-TAMRA is readily observed in the presence of Nsp15 (+Nsp15) as a singly charged ion at $m/z$ 1463.42, but is undetectable in the absence of enzyme (-Nsp15). NL stands for normalized level. **b** MS spectrum confirms the identity of 5'-HO-AA-TAMRA and does not detect the presence of alternative 5'-cleavage products. **c** Extracted ion chromatograms of the 5'-fluorescein labeled (Fl) Fl-AAAU cleavage product. Fl-AAAU is only detected in the presence of Nsp15. **d** MS spectrum confirms the identity of Fl-AAAU. The 3'-product is observed primarily as a doubly charged ion at $m/z$ 914.14, which corresponds to a 2'3'-cP terminated moiety. A doubly charged ion at $m/z$ 923.14 is also present and corresponds to a 3'-P terminated species. MSMS spectra of $m/z$ 1463.14 and $m/z$ 914.14 unambiguously confirm the identity of these ions (Fig. S5) **e** Extracted ion chromatograms of the 5'-Fl-AAAU-2'3'-cP cleavage product (top) and the 5'-Fl-AAAU-3'-P cleavage product (bottom) set to the same scale to demonstrate that under the conditions employed, the majority of the cleavage product is terminated by a cyclic phosphate. The graphical representation of the areas under the curve of the extracted ion chromatograms shows that approximately 80% of the cleavage product is the cyclic phosphate (assuming similar ionization efficiencies of the two species). **f** Cartoon schematic of RNA cleavage by the uridine-specific Nsp15 endoribonuclease. Nsp15 catalyzes a transesterification reaction that results in 2'3'-cyclic phosphate (2'3'-cP) and 5'-hydroxyl (5'-OH) RNA ends. Nsp15 can further catalyze a subsequent hydrolysis reaction resulting in the conversion of the 2'3'-cyclic phosphate to a 3'-phosphate (3'-P), however, the rate of hydrolysis is slow. R denotes the 5' and 3'-ends of the phosphodiester backbone.

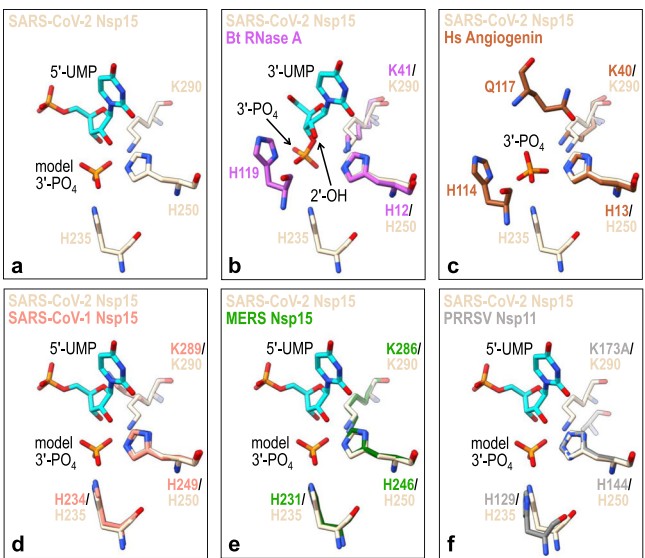

**Fig. 4 SARS-CoV-2 Nsp15 active site superimposition with metal-independent endoribonucleases. a** SARS-CoV-2 Nsp15 catalytic residues are shown as sticks (PDB ID 7K0R; beige) along with its bound 5'-UMP nucleotide (cyan) and model 3'-phosphate (3'-PO₄). **b** *Bos taurus* (Bt) RNase A (PDB ID: 1O0N; magenta) bound to 3'-UMP (cyan)[25], **c** *Homo sapiens* (Hs) angiogenin (PDB ID: 1HBY; brown) bound to 3'-phosphate (3'-PO₄)[44], **d** SARS-CoV-1 Nsp15 (PDB ID: 2H85; orange)[45], **e** MERS Nsp15 (PDB ID: 5YVD; green)[4], and **f** PRRSV Nsp11 (PDB ID: 5EYI; gray)[46] are superimposed with the SARS-CoV-2 Nsp15 active site.

hydrolysis of the cyclic phosphate. This difference in Nsp15 active site architecture likely slows the hydrolysis reaction leading to an accumulation of 2'3'-cP products. This is reminiscent of the active site of a related nuclease angiogenin[26]. The active site of angiogenin is partially blocked by a glutamine residue that is not found in RNase A (Fig. 4c)[27]. The glutamine residue alters the active site geometry in such a way that cleavage is not as efficient as RNase A and it primarily produces 2'3'-cP products. Similarities between the SARS-CoV-2 Nsp15 active site and the catalytic residues of SARS-CoV-1 Nsp15, MERS Nsp15, and PRRVS Nsp11 suggest a common mechanism for RNA cleavage by transesterification (Fig. 4d–f).

**Cryo-EM reveals endoU dynamics.** In addition to determining the structure of Nsp15 in the presence of UTP, we also determined several cryo-EM reconstructions of Nsp15 in the absence of ligand at resolutions ranging from 2.9 to 3.3 Å (Supplementary Figs. 8–10). First, we vitrified the H235A-Nsp15 mutant, which

cannot cleave RNA, in the presence of a 21-nucleotide single-strand RNA (SS-RNA) substrate in an attempt to solve the structure of Nsp15 bound to RNA. We processed these data with and without imposing D3 symmetry, but did not observe any asymmetric states or the presence of RNA. Overall, the cryo-EM reconstruction of apo H235A-Nsp15 is very similar to the uridine-bound reconstruction with one notable difference, an observed loss in the local resolution of the endoU domain (Supplementary Fig. 8). The ND and MD domains could be modeled into the apo H235A-Nsp15 map with high confidence and the structure of these domains is very similar to the UMP-bound model (RMSD of 0.21 Å for the ND and MD domains). The poor density of the endoU domain prevented us from building an accurate model of the domain. One possibility for this loss in resolution is the presence of heterogeneity in the conformation of the endoU domain. We carried out 3D variability analysis to model the conformational landscape of the endoU domain present in our cryo-EM data[28]. This analysis revealed multiple conformational states of the endoU domain with respect to the remaining Nsp15 hexamer (Fig. 5a and Supplementary Movie 2). The conformational variability of the endoU domain is restrained, but suggests that in the absence of a ligand the endoU domain wobbles towards the central channel with respect to the ND and MD domains and is not locked into a fixed position.

In order to confirm that the observed wobbling of the endoU domain was not due to the H235A active site mutation or the presence of RNA when vitrifying the sample, we determined two additional cryo-EM structures of wt-Nsp15 and Nsp15 H235A vitrified in the absence of any ligand. Both of these reconstructions also display a loss in the local resolution of the endoU domain in comparison with UTP-bound Nsp15 (Supplementary Figs. 9 and 10). We repeated 3D variability analysis on these datasets and observed similar conformational landscapes of the endoU domain (Fig. 5a and Supplementary Movies 3 and 4). Collectively, the three apo Nsp15 cryo-EM reconstructions reveal that the endoU domain of Nsp15 is dynamic and samples multiple conformations in its ligand-free state.

To gain additional insight into the conformational variability of the endoU domain we used rigid-body refinement to dock the Nsp15 hexamer into the 20 individual reconstructions derived from 3D variability analysis[28]. The individual domains of Nsp15 were treated as separate rigid bodies. The superposition of the coordinates reveals that in contrast to the minor movements observed in the ND and MD, the endoU domain samples numerous conformational states in the absence of ligand (Fig. 5b, c) The MD and endoU domains are connected by a linker whose sequence is not well conserved across Nsp15 family members (Fig. 5c and Supplementary Fig. 5), but is likely an important hinge point for regulating the motion of the endoU domain.

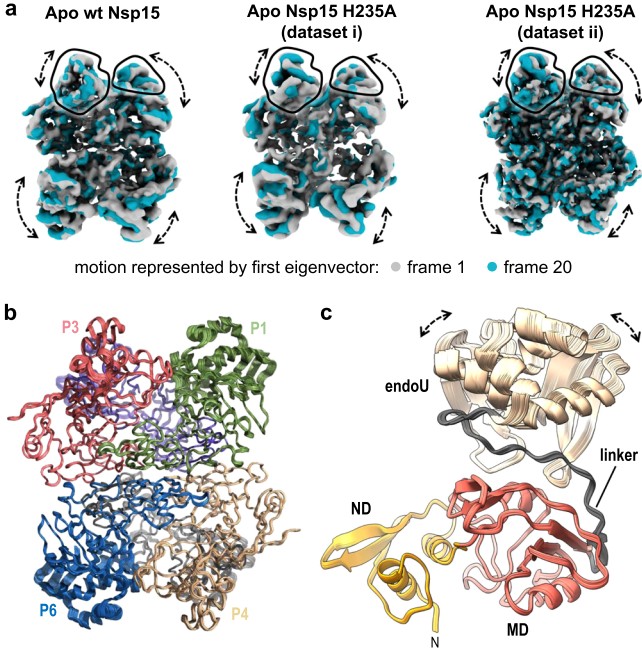

Apo wt Nsp15 Apo Nsp15 H235A (dataset i) Apo Nsp15 H235A (dataset ii)

motion represented by first eigenvector: ● frame 1 ● frame 20

**Fig. 5 3D Variability of the endoU Domain. a** 3D variability analysis[28] of cryo-EM reconstructions of Nsp15 apo-states. Frame 1 (gray) and frame 20 (blue) of a 20-frame set describing the motion by the first eigenvector of the cryo-EM reconstructions. Black outlines demarcate the unique views of the endoU domains and dotted arrows demarcate regions of conformational heterogeneity. **b** Superposition of rigid-body docked models of the Nsp15 hexamer into the Apo Nsp15 H235A (dataset ii) reconstructions derived from 3D variability analysis. **c** View of the individual P1 protomer from (**b**) illustrating the conformational dynamics of the endoU domain. The linker between the endoU and MD is colored in gray.

While the atomic models of Nsp15 derived from cryo-EM and X-ray crystallography are in excellent agreement with one another, the apo Nsp15 cryo-EM data reveal that the endoU domain is dynamic. This discovery of endoU dynamics reinforces the power of combining multiple structural biology techniques in order to capture different conformational states of dynamic macromolecules.

**UTP binding fixes the position of the endoU domain**. Finally, we used molecular dynamics simulations to confirm the observed wobbling of the endoU domain in the absence of ligand. We looked at the dynamics of the Nsp15 monomer and Nsp15 hexamer in the presence and absence of 5′-UMP. We established the stability of the Nsp15 monomer and hexamer by determining the root mean square deviations (RMSD) from molecular dynamics trajectories following the convergence of the RMSD values in the last 100 ns of the simulations (Supplementary Fig. 11). There is a significant difference in the RMSDs of the monomer compared to the averaged values of individual monomers from the hexamer, both in the absence (2.0 vs 1.3 Å) and presence (2.3 vs 1.4 Å) of 5′-UMP. Thus, the hexamer provides stability for the individual Nsp15 protomers and this stability likely contributes to the requirement of Nsp15 oligomerization for RNA cleavage activity.

Next, we determined the impact of 5′-UMP binding on the dynamics of the monomer and hexamer. Binding of 5′-UMP does not result in large changes in the thermal fluctuations, but results in differences with the dynamic correlation (Supplementary Fig. 11). Dynamic correlation reflects the association between the distances of the alpha-carbons across the entire protein backbone.

In the absence of 5′-UMP we observe correlations within the residues of individual domains of Nsp15, but there is little correlation among the three distinct domains. This suggests the ND, MD, and endoU domains are largely moving independently of one another in the absence of a ligand. In contrast, we observe an increase in the correlated motions of both the residues within the endoU domains across the hexamer and among the ND, MD, and endoU domains in the presence of 5′-UMP (Fig. 6a and Supplementary Table 2). This observation is in-line with 5′-UMP-induced ordering of the Nsp15 endoU domain and a previous report that the addition of mononucleotide and dinucleotide analogs significantly increases Nsp15 stability, by differential scanning fluorometry[24]. Moreover, a deposited crystal structure of Nsp15 with nothing bound in the active site has higher B-factors in the endoU domain in comparison with the UMP-bound Nsp15 structure (Supplementary Fig. 12). Finally, we also calculated the average distance between important active site residues within a protomer from the replicates of all four MD-derived models (Supplementary Table 2). We do not observe any significant deviations of these distances in the presence or absence of ligand. This suggests that the active site within the endoU remains ordered as a rigid body as it wobbles with respect to the ND and MD domains. In conclusion, both our cryo-EM analysis and molecular dynamics simulations suggest that 5′-UMP binding locks the endoU domain into a fixed position that correlates its motion with the rest of the Nsp15 higher-order assembly.

## Discussion

Here we present a cryo-EM-derived atomic model of the hexameric endonuclease Nsp15 from SARS-CoV-2. In the presence of 5′-UMP, Nsp15 has an ordered active site that draws many parallels with the well-studied RNase A, which utilizes an elegant acid–base reaction mechanism to catalyze a two-step reaction of transesterification and hydrolysis. Mass spectrometry revealed that analogous to RNase A, Nsp15 can catalyze both reaction steps; however, we observed a significant accumulation of the 2′3′-cP product from the transesterification reaction indicating that hydrolysis is slow. Based on the comparison of the active sites of Nsp15 and RNase A, this difference can be attributed to an alternative location of H235 in the Nsp15 active site. While the nature of the 3′-end of the cleavage product can have important implications for downstream RNA processing and/or function[26], it seems unlikely that the identity of the 3′-end is critical for viral pathogenesis. Nsp15, with a presumptive role in regulating the length of poly(U) tails at the 5′-end of the negative-strand (Supplementary Fig. 1), generates a shortened negative-strand with a 5′-hydroxyl and a range of short U-ended oligonucleotide products with a mixture of 2′3′-cP and 3′-P ends[8].

The 5′-UMP-bound cryo-EM structure is similar to existing Nsp15 crystal structures, however, the apo cryo-EM reconstructions revealed structural heterogeneity within the endoU nuclease domain. This heterogeneity demonstrates that the endoU domain wobbles with respect to the N-terminal half of Nsp15 in the absence of RNA ligand. Molecular dynamics confirmed that in the absence of ligand the endoU domain moves independent of the rest of the molecule. In contrast, in the presence of 5′-UMP the endoU domain is locked down and its movement is correlated with the N-terminal half of the protein (Fig. 6b). While more work is needed to establish the functional significance of the endoU wobble, there are several possibilities as to why this could be critical for Nsp15 function. Flexible active sites that become stiff upon engaging substrates have been shown to promote catalysis by reducing the substrate-binding energy[29]. Another possibility is that the wobbling of the endoU domain is important for

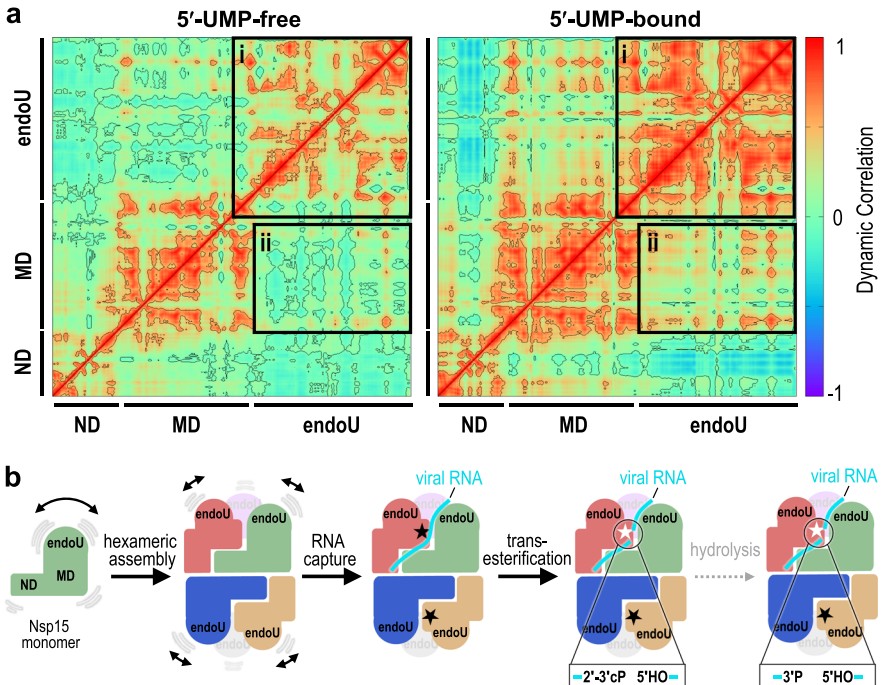

**Fig. 6 Molecular dynamics reveal UMP locks the endoU domain. a** Dynamical cross-correlation matrix (DCCM) analysis of a 5′-UMP-bound protomer and its 5′-UMP-free form from hexamer systems. Both inter- and intra-domain positive correlations are significantly enhanced due to 5′-UMP binding. The correlations were calculated from the equally spaced 100-configurations extracted from the last 500 ns of each simulation. Black boxes highlight the positively correlated motions between the endoU domains within the hexameric assembly (box i) and the endoU domain relative to the N-terminal half of the protomer (box ii). **b** Cartoon illustrating the dynamics of Nsp15 and the mechanism of RNA cleavage. Monomeric Nsp15 is stabilized by the formation of a hexameric assembly. In the absence of ligand the endoU domain exhibits conformational dynamics; however, upon binding to the RNA the endoU domain becomes stabilized. The fixed endoU active site cleaves viral RNA and primarily generates products with a cyclic phosphate.

Nsp15 to accommodate different RNA substrates. Recent work suggests that in addition to cleaving the 5′-end of the negative-strand, Nsp15 can cleave multiple sites within the viral RNA to prevent activation of host anti-viral sensors[8,13]. Finally, wobbling of the endoU domain may provide Nsp15 with a mechanism of allosteric communication between the six individual protomers. While we only observed Nsp15 particles with 6-fold symmetry, we hypothesize that binding of large RNA substrates will lead to the formation of asymmetric states that rely on coordination between the protomers to bind and process RNA. This is further supported by the free energy estimations from molecular dynamics simulations, which revealed that 5′-UMP initially binds all six active sites, but only remains stably associated with a single protomer (Supplementary Table 3).

In summary, our cryo-EM reconstructions, biochemistry, and molecular dynamics reveal critical insight into how SARS-CoV-2 Nsp15 processes viral RNA. The series of structures presented in this manuscript can be used to aid in the development of effective inhibitors, which are urgently needed as reported cases of Covid-19 continue to rise. Nucleotide analogs have shown great promise as viral inhibitors against SARS-CoV-2, thus uridine derivatives such as Tipiracil, that could compete for binding to the Nsp15 active site should continue to be explored as new therapeutic targets[24]. Moreover, recent in silico-based approaches have identified potential Nsp15 inhibitors that await structural and biochemical validation[30,31].

## Methods

**Nsp15 construct design**. The nucleotide sequence for the Tobacco Etch Virus (TEV) recognition site was added upstream of full-length SARS-CoV-2 Nsp15 that was codon optimized for *Escherichia coli* expression (Supplementary Table 4). The TEV-Nsp15 construct was inserted into the bacterial expression vector pET-14b in frame with the N-terminal 6x His tag and thrombin cleavage site to create the 6x-His-Thrombin-TEV wt-Nsp15 construct. Nsp15 catalytic-deficient variants H235A and H250A were created using the wt-Nsp15 template (Supplementary Table 4). All plasmids were verified by DNA sequencing (GeneWiz).

**Protein expression of Nsp15 variants**. Wt-Nsp15 was overproduced in *E. coli* C41 (DE3) competent cells cultured in Terrific Broth supplemented with 100 μg/mL ampicillin. Transformed cell cultures were grown to an optical density (600 nm) of 1.0 prior to their induction with 0.2 mM Isopropyl β-D-1-thiogalactopyranoside (IPTG) and incubated for 3 h at 37 °C. Harvested cells were stored at −80 °C until needed. Catalytic-deficient Nsp15 variants (H235A or H250A) were overexpressed in *E. coli* Rosetta (DE3) pLacI competent cells in Terrific Broth supplemented with 100 μg/mL ampicillin and 25 μg/mL chloramphenicol. Transformed cell cultures were grown to an optical density (600 nm) of 1.0. Cell cultures were stored at 4 °C for 1 h prior to their induction with 0.2 mM IPTG and overnight incubation at 16 °C. Harvested cells were stored at −80 °C until needed.

**Protein purification**. Cells were resuspended in lysis buffer (50 mM Tris pH 8.0, 500 mM NaCl, 5% glycerol, 5 mM β-ME, 5 mM imidazole) supplemented with cOmplete EDTA-free protease inhibitor tablets (Roche) and disrupted by sonication. The lysate was clarified by centrifugation at 26,915 g for 50 min at 4 °C and then incubated with TALON metal affinity resin (Clontech). His-Thrombin-TEV-Nsp15 variants were eluted with 250 mM imidazole and further purified by gel filtration on a Superdex-200 column (GE Healthcare) in SEC buffer (20 mM HEPES pH 7.5, 150 mM NaCl, 5 mM MnCl₂, 5 mM β-ME). To remove the N-terminal 6x His-tag from His-Thrombin-TEV-Nsp15, variants were incubated with thrombin (Sigma) at room temperature in Thrombin Cleavage Buffer (50 mM Tris pH 8.0, 150 mM NaCl, 5% glycerol, 2 mM β-ME, 2 mM CaCl₂) for 3 h. Thrombin cleavage was quenched by the addition of 1 mM PMSF (phenylmethylsulfonyl fluoride). Proteolytic cleavage reactions were incubated with TALON metal affinity resin and tagless protein was eluted in batch and resolved by gel filtration using a Superdex-200 column equilibrated with SEC buffer.

**Preparation of Nsp15 cryo-EM specimens**. Purified wt-Nsp15 and catalytic-deficient Nsp15 H235A were resolved over a Superdex-200 (GE Healthcare) gel filtration column using SEC buffer to isolate the hexamer. Prior to grid preparation

Nsp15 variants were diluted into low-salt buffer (20 mM HEPES pH 7.5, 100 mM NaCl, 5 mM MnCl$_2$, 5 mM β-ME). Nsp15 variants (0.75 μM) were incubated in the presence of excess 8 μM 22-nucleotide uridine-rich single-stranded RNA (UUUAGGUUUUACCAACUGCGGC/36-TAMSp/) (H235A-APO, dataset i), in the absence of ligand (H235A-APO, dataset ii), in the presence of excess 333 μM 22-nucleotide single-stranded RNA (SS-RNA: 5′-ACGUACGCGGAAUA-CUUCGAA-TAMRA-3′) (wt-APO), and in the presence of excess 2 mM uridine 5′-triphosphate (wt-UTP bound) for 1 h at 4 °C. UltrAuFoil R1.2/1.3 300 mesh gold grids (Quantifoil) were rendered hydrophilic using the Tergeo plasma cleaner (Pie Scientific). Protein mixtures (5 μL) were deposited onto the grids and blotted for 3 s using an Automatic Plunge Freezer (Leica).

**Data acquisition and image processing**. Images of Nsp15 variants were collected on a Talos Arctica electron microscope (Thermo Fischer Scientific) operated at 200 keV with no energy filter and equipped with a K2 Summit direct detection camera (Gatan). Movies were recorded in counting mode at a nominal magnification of 45,000x and 36,000x corresponding to 0.932 and 1.187 Å/pixel, respectively (see Table 1). Total exposure time was 8.4 s at a flux of 6.54 e$^-$/Å$^2$/s resulting in a fluence of ~54 e$^-$/Å$^2$, distributed over 60 frames. Images were recorded across an underfocus range of −1.0 to −2.5 μm.

Beam-induced motion and drift were corrected using MotionCor2[32] and aligned dose-weighted images were used to calculate CTF parameters using CTFFIND4[33]. CryoSPARC v2[34] was used in all subsequent image processing. Particles were selected by template-based particle picking, downsampled by a factor of 4, extracted with a box size of 64, and subjected to an initial round of 2D classification. Full resolution particle projections from "good" classes were re-extracted using a box size of 256. Ab initio reconstruction with three classes was used to generate initial models. Three independent 3D refinement cycles were performed while applying C1, C3, and D3 symmetry, respectively. Visual examination of the refined maps did not reveal significant differences between asymmetric units. For this reason, refined maps with D3 symmetry were used for all subsequent model building and analysis. Local resolution was calculated using cryoSPARC's own implementation. Particles from each refinement were then post-processed using per-particle motion correction and global CTF refinement before undergoing a subsequent iteration of non-uniform refinement. Local refinement of the asymmetric unit of the map was performed by masking the asymmetric unit. The mask was padded by 5 pixels with a soft mask extended by 3 pixels. Each dataset was D3 symmetry expanded and locally refined with the fulcrum point defined as the center of the mask. A 3D variability analysis was performed using cryoSPARC's own implementation[28]. The full mask from 3D refinement of each respective dataset was used, along with a D3 symmetry expanded dataset. A 5 Å filter resolution and 20 Å high pass resolution were used to capture movements of each asymmetric unit. Maps were visualized using the simple 3D variability display job with a total of 20 frames. Superposition of the model versus high-resolution crystal structures revealed an error in the voxel size. Therefore, the map was rescaled by 0.9647 to optimize the RMSD fit of the ND to crystal structures PDB ID 6WLC and 6X4I.

**Model building**. The SARS-CoV-2 Nsp15 crystal structure bound to 5′-UMP (PDB ID: 6WLC) was fit into the UTP-bound cryo-EM reconstruction using rigid-body docking in Phenix[35]. The nucleotide density for the uracil base, ribose sugar, and α-phosphate is visible, however, the density for the β- and γ-phosphates of the UTP molecules was ambiguous. For this reason, we modeled 5′-UMP. Additional density was also observed adjacent to the nucleotide within the active site. We modeled phosphate into this density, however, the identity remains unknown. Fit was improved using a combination of rigid-body and real-space refinement in Phenix[35] combined with iterative rounds of building in COOT[36]. Hydrogen bond restraints were included for the UMP ligand to stabilize refinement in the poorly resolved ligand density. Molprobity[37] was used to evaluate the model and the model statistics are listed in Table 1. Figures and videos were prepared with PyMOL, Chimera, ChimeraX, and CCP4MG[38–41]. Model building for the apo-state cryo-EM reconstructions were not performed due to poor endoU domain density caused by conformational heterogeneity.

**Nsp15 endoribonuclease assay**. Real-time Nsp15 RNA cleavage was monitored as previously described[4] with minor modifications. The 5′-fluorescein (FI) label fused to the FRET RNA substrate is quenched by its 3′-TAMRA label (5′-FI-AAAUAA-TAMRA-3′). The FRET RNA substrate (0.8 μM) was incubated with a constant amount of tagless Nsp15 variant (2.5 nM) in RNA cleavage buffer (20 mM HEPES pH 7.5, 75 mM NaCl, 5 mM MnCl$_2$, 5 mM β-ME) at 25 °C for 60 min. RNA cleavage was measured as an increase in fluorescein fluorescence. Fluorescence was measured every 2.5 min using a POLARstar Omega plate reader (BMG Labtech) set to excitation and emission wavelengths of 485 ± 12 nm and 520 nm, respectively. Three technical replicates were performed to calculate the mean, standard deviation, and two-tailed Student's $t$ tests.

**Mass spectrometry analysis of RNA cleavage products**. To analyze the masses of the RNA cleavage products produced by Nsp15, the FRET RNA substrate (0.8 μM) was incubated in the absence and presence of tagless Nsp15 variant (2.5 nM) in RNA Cleavage Buffer at 25 °C for 30 min. Mass spectrometry was performed essentially as previously described[42] with the following modifications. Buffer A was 400 mM hexafluoro-2-propanol, 3 mM triethylamine (pH 7.0) and buffer B was methanol. Additionally, parallel reaction monitoring (PRM) analyses were included in the MS analyses with included masses of $m/z$ 914.14; 923.14; 1463.42.

**Molecular dynamics**. Initial monomer structures were selected from the crystal structure of the SARS-CoV-2 Nsp15 endoribonuclease bound to 5′-UMP (PDB ID: 6WLC). Four simulation systems were constructed, (a) a ligand-free monomer, (b) a 5′-UMP -bound monomer, (c) a ligand-free hexamer, and (d) a 5′-UMP-bound hexamer (with six 5′-UMP molecules). After introducing protons, each structure was solvated in a box of water, counter ions were added, and additional Na$^+$ and Cl$^-$ ions were placed so that the salt concentration was 100 mM. There were 60/48/26628, 62/48/26615, 230/158/87369, and 242/158/87291 Na+ ions/Cl− ions/water molecules present in systems (a), (b), (c), and (d) described above. The closest box boundary is at least 15 Å away from any protein atom in the rectangular boxes with the initial box dimensions of 95.30 Å × 99.18 Å ×91.49 Å and 134.55 Å × 153.98 Å× 145.07 Å for monomer and hexamer systems, respectively. The charges for 5′-UMP were generated at the 6−31 g*/B3LYP level with the FF14SB force field for amino acids. The protonation state of histidine residues was selected by manually inspecting the environment of each residue and all but H250 ware protonated at the ε− positions. To facilitate the H-bond between S294 (carbonyl oxygen) and H250 (Nδ), H250 was protonated at the δ− position. After proper equilibration of each system over 30 ns under various conditions, the CUDA implementation of the PMEMD module of Amber.18[43] was used to simulate unconstrained dynamics for 500 ns for monomer systems at 2 fs time step and 300 K under constant pressure. The particle mesh Ewald method was used in dealing with long range Coulomb and van der Waals interactions. The hexamer trajectories were extended to a microsecond. For each system, two additional simulations were performed. The starting structures of these runs were selected from the 30 and 40 ns conformations of the primary simulation with the randomized initial velocities to simulate alternate trajectories. The MMGBSA module of Amber.18 was implemented in free energy estimations with the selection of 0.15 M salt concentration and the default parameters (IGB = 5) in the Amber module.

**Reporting summary**. Further information on research design is available in the Nature Research Reporting Summary linked to this article.

## Data availability
The cryo-EM maps and atomic coordinates for Nsp15 have been deposited in the Electron Microscopy Data Bank and PDB under the following access numbers: EMDB: 22610, 22611, 22612, and 22613 and PDB: 7K0R. Other data are available from the corresponding authors upon reasonable request. Source data are provided with this paper.

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

## Acknowledgements

We thank Dr. Traci Hall and Dr. Joseph Rodriguez for their critical reading of this manuscript. We would like to thank all the members of the Molecular Microscopy Consortium at the NIEHS for their help with cryo-EM data collection and processing, along with all of the members of the Mass Spectrometry Research and Support Group at the NIEHS for their help with mass spectrometry data collection and analysis. We are also grateful to Dr. Oliver Clarke for his support with cryo-EM data processing using cryoSPARC. This work was supported by the US National Institutes of Health Intramural Research Program; US National Institute of Environmental Health Sciences (NIEHS) (ZIA ES103247 to R.E.S., Z01 ES043010 to L.P., 1ZI CES102488 to J.G. W., 1ZI CES103206 to L.J.D., and ZIC ES103326 to M.J.B.). This work was also supported by the NIH Intramural Targeted Anti-COVID-19(ITAC) Program funded by the National Institute of Allergy and Infectious Diseases. We would also like to acknowledge the NIH-Oxford-Cambridge Scholars Program for support of J.G. as a graduate student.

## Author contributions

R.E.S. and M.C.P. conceived and designed the study. M.C.P., L.C.D., J.M.K., S.K., C.K.H., J.G., and Z.D.S. performed cryo-EM analysis and structure determination and refinement. L.C.D, A.H., V.D., and M.B. collected and processed cryo-EM data. M.N.F. purified all recombinant proteins. S.K. expressed recombinant proteins. C.K.H. and M.S. generated recombinant DNA and other materials. J.G.W. and L.J.D. processed and analyzed mass spectrometry data. L.P. performed molecular dynamics simulations. M.C.P., L.B.D., M.N.F., C.K.H., J.G., J.G.W., L.J.D., and R.E.S. prepared the figures. M.C.P., M.N.F., and R.E.S. wrote and revised the manuscript.

## Competing interests

The authors declare no competing interests.
