## [Peer Review File · Nature Communications]

REVIEWER COMMENTS

Reviewer #1 (Remarks to the Author):

Coronavirus nsp15 is a uridine specific endoribonuclease, which processes conservative structure and function. Here, Pillon et al report the first series of cryo-EM reconstructions of SARS-CoV-2 Nsp15, in both apo and UTP-bound states. Their results showed that nsp15 participate in the binding and cleavage of RNA through the swing between monomers, and perform other activities. The findings will be of interest to researchers studying coronavirus replication. There are, however, several deficiencies with the submission, as itemized below.

In this study, the author analyzed four SARS-CoV-2 nsp15 hexamer structures by cryo-electron microscopy technology, which were shown in Figs S2, S6, S7 and S8. Are there differences in these structures?

Furthermore, whether the structure of the complex of Nsp15 and 5'-UMP has been resolved, if so, please show the electron density of 5'-UMP.

If the conformational change of SARS-CoV-2 nsp15 is related to its function, is there an increase in the proportion of particles with significant conformational changes in Fig S6 where the nucleic acid substrate exists? In Fig 6, please show the structure of 5'-UMP-bound protomer and its 5'-UMP-free form from hexamer systems, and compare the structures. At the same time, please mark the RNA binding channel in the nsp15 hexamer structure.

Finally, the core of this research is the discovery of dynamic changes in the SARS-CoV-2 nsp15 hexamer (Fig S6-S8). However, it is not clear dynamic changes of these maps from which particle conformations. Please show the electron density of these different particle conformations with conformation changes in Fig S6-S8. In addition, please explain the correlation between these conformational changes and Nsp15' function, and how nsp15 can perform the function of endoribonuclease through these conformational changes. What factors drive these changes?

Other comments

1. Fig1 is just a summary of early literature reports, it should be added to supplementary materials (lines 52, 75)

2. Following 3D classification, the particles converged into a single prominent class with a resolution of 3.36 Å. However, the overall resolution is 3.38 Å (Fig. S2D), 3.72 Å (Fig. S2G) and 3.36 Å (Fig. S2G). Please explain what the difference between these different resolutions means. (lines 106-107); Similar to Fig S6

3. Coronavirus Nsp15 is a protein with conserved structure and function. It is recommended to compare SARS-CoV-2 Nsp15 with the resolved structure of coronaviruses nsp15, and provide RMSD value (line 119). Besides, SARS-Cov-2 is SARS-CoV-2

4. Lines 127: It seems that 5'-UMP has not observed the corresponding electron density through cryo-electron microscopy, but the analysis of the effect of 5'-UMP and SARS-CoV-2 nsp15 active site is completed through molecular docking. Furthermore, the analysis of the active sites of 5'-UMP and Nsp15 did not clarify new ideas compared with previous literature reports. Besides, please compare the difference between SARS-CoV nsp15 and SARS-CoV-2 nsp15 nuclease activity

5. Line226 Fig. 3F is Fig. 4F

Compared with the structure of RNase A active sites, it is found that H235 from Nsp15 and H119 from RNase A do not superimpose, are the structures of other coronavirus Nsp15 similar? Please further explain how the weakening of nsp15 activity affects the invasion and proliferation of the virus itself.

6. Why are there differences in the Protein expression between Wt-Nsp15 and Catalytic-deficient Nsp15 variants (H235A or H250A)? (lines 351-361)

7. Through cryo-electron microscopic analysis, the author found that there is a swing between the SARS-CoV-2 nsp15 monomer molecules, and believes that this swing may be involved in the binding of RNA. Generally, RNA is negatively charged. Is the RNA-binding region between nsp15 monomers identified in the article positively charged? Furthermore, the observed mutual movement between the monomers is very slight. Is there room for single-stranded RNA? In addition, nsp15 participates in the formation of the replication complex in the process of coronavirus replication. Does the movement between its monomers facilitate the interaction with other replication-related proteins?

At present, through data analysis, is it possible to obtain the structure of different conformational states of nsp15 swinging between monomers? It is not clear that how the large swing is observed in the movies. If the resolution is very low, by continuously displaying maps of different conformations, does it truly represent the existing conformation changes?

8. Whether coronavirus nsp15 can non-specifically cut the viral genome RNA and host RNA, if so, the loss of nsp15 activity does not seem to improve the replication ability of the virus, please discuss.

9. After the loss of coronavirus Nsp15 activity, the virus still seems to be able to replicate. Therefore, nsp15 may be a target for vaccine development, not a target for drugs.

Reviewer #2 (Remarks to the Author):

In this manuscript Pillon, Frazier and Dillard et al. determine cryoEM single particle structures of the SARS-CoV-2 endoribonuclease Nsp15, study its enzymatic mechanism by substrate cleavage assays and provide insights into its dynamics and substrate binding properties.

The authors show that Nsp15 in solution is organized into a hexamer, a dimer of trimers. The structure bound to UMP reveals details of the interaction with the substrate which allows the authors to speculate on the enzymatic mechanism. They utilize substrate cleavage assays followed by mass spectrometry to interrogate the enzymatic reaction, revealing the accumulation of 2'3'-cyclic phosphate cleavage products. The catalytic EndoU domain is more flexible in the absence of a substrate in cryoEM data-based structural variability analysis and MD simulations. This most likely helps efficient substrate interaction. The results presented here provide crucial novel insights into the mechanism of this key viral endonuclease, paving the way for potential future drug development and a deeper understanding of the molecular basis for SARS-CoV-2's life cycle and host interactions.

The study is well designed, the included data are of very high quality and the conclusions reached are supported by the data provided. The manuscript is very well written and the figures are informative and well structured.

Before publication the authors should address a few points outlined in the following.

Major points:

1) The authors should provide evidence for the correlation between their structural models and the respective 3D reconstructions as a measure of model confidence. This is typically done by a flavor of the CCmap vs model, the FSCmap vs model plot and cut off resolution at 0.5 and 0.143.

2) Do the authors see differences in the arrangements of the endoU active site of Nsp15 in the UMP-bound, apo and mutant 3D reconstructions they calculated?

3) An additional figure panel illustrating the direct comparison between the Angiogenin active site with the active site of the UMP-bound state (and potentially again the RNase A active site) would be of great help for readers.

4) How does the recent x-ray crystallographic structure(s) compare to the cryoEM structures determined here? Do the B-factors of the model based on the x-ray crystallographic data correlate with the domain movements/flexibility the authors observe in their variability analysis and MD analysis?

5) Do the authors expect UMP to function as an inhibitor of the cleavage reaction? Is there a straight forward way to test this? Data down these lines would further strengthen the manuscript significantly in light of future structure-guided drug development.

6) How do the crucial active site residues the authors identify behave in their MD simulations? How do their distances to each other and the UMP moiety behave over the time course of the simulation?

Minor points:

A recent preprint published on bioRxiv (<https://doi.org/10.1101/2020.06.26.173872>) concludes that Tipiracil inhibits SARS-CoV-2 Nsp15 by binding to the active site. How does the Tipiracil binding mode proposed in this preprint relate to the authors conclusions on the enzymatic mechanism of Nsp15?

Fig.3 C The authors might want to refrain from indicating potential hydrogen bonds by dotted lines in the figures and rather describe them as such in the text/figure legend. While these contacts are likely the overall estimated resolution of the reconstruction at 3.38 Å does not support strong graphic conclusions of this kind.

Fig. 2. B) The authors might want to include labels for the domain arrangement in at least one of the protomers of the hexamer again for ease of understanding the multimeric arrangement of the complex

Is the Arctica the authors used equipped with a post-column energy filter before the K2 camera (and did they use it)? If so the authors should include this information in the methods section

Line 214 "...m/z/ 923.14..." should be replaced by "m/z 923.14"

Line 218 "... (Fig. 3E)..." should be replaced by "(Fig. 4E)"

Reviewer #3 (Remarks to the Author):

In this manuscript, the authors reported several cryo-EM structures of SARS-CoV-2 nsp15 in its apo form as well as in complex with UMP. In my opinion, the authors draw two major conclusions based on these structures. In the first one, they proposed the critical role of His235 and His250 in chemical catalysis of the endonuclease cleavage. Specifically, they suggest the mechanism, in which His250 serves as the general base to activate the attacking nucleophile in hydrolysis. Their subsequent biochemical assays on the H250A mutant validated the critical role of this residue. In the second conclusion, based on the analysis of several cryo-EM datasets and MD simulations, the authors showed that the binding of UMP can greatly reduce the conformational flexibility and lock the EndoU domain. Overall, this manuscript addressed an interesting topic, and SARS-CoV-2 nsp15 could serve as a novel drug target for COVID-19. However, I am not fully convinced by their first conclusion, though their second conclusion is interesting. See below for my comments:

1. In this work, the authors solve the structure of nsp15 in complex with UMP, rather than with a RNA chain. Similar structures have already been reported in Ref. 11 solved via X-ray crystallography. The critical roles of residues His235 and His250 have also been identified in Ref 11. This reduces the novelty of the current work.
2. The authors proposed a catalytic mechanism for the endonuclease RNA cleavage. In this mechanism, His150 serves as the general base to activate 2'OH as the attacking nucleophile. However, this is only based on a modelled structure by simply introducing a 3'-PO₄ into their nsp15-UMP structure (see Fig. 3C). This is not convincing to me. In my opinion, the binding of RNA chain will impact the orientation of its nucleotide in the active site, which could lead to a different conformation from the UMP+3'PO₄. Therefore, I would suggest that the authors should attempt to model a short RNA chain in complex with nsp15 to further validate their proposed mechanism.
3. The MD simulation part is interesting. However, a lot of details of their simulation set-up are missing (see below for some examples):
 - The number of waters in their simulation box
 - The type and size of the box
 - The number of counter ions added
 - How to deal with coulomb and vdw interactions?
 - How to choose the protonated states of protein residues.
 - Parameters for the MMGBSA to compute free energy.
4. My understanding is that the authors only perform a single MD simulation for each system. The authors need to perform additional MD simulations to show that their reported results are statistically significant.

NCOMMS-20-35675 Rebuttal

Reviewer comments (*blue italics*) and our response (black)

Reviewer #1 (Remarks to the Author):

Coronavirus nsp15 is a uridine specific endoribonuclease, which processes conservative structure and function. Here, Pillon et al report the first series of cryo-EM reconstructions of SARS-CoV-2 Nsp15, in both apo and UTP-bound states. Their results showed that nsp15 participate in the binding and cleavage of RNA through the swing between monomers, and perform other activities. The findings will be of interest to researchers studying coronavirus replication. There are, however, several deficiencies with the submission, as itemized below.

We thank the reviewer for their support of our manuscript and their constructive comments for improvement. Please note that we broke the reviewer comments up into numbered bullet points so that we could individually address each concern.

Major Points

1.1 In this study, the author analyzed four SARS-CoV-2 nsp15 hexamer structures by cryo-electron microscopy technology, which were shown in Figs S2, S6, S7 and S8. Are there differences in these structures?

The 4 cryo-EM reconstructions are all very similar, with the exception of the variability in the endoU domain. We performed rigid body docking of the Nsp15 N-terminal domain (ND), middle domain (MD), and endoU domain into each cryo-EM reconstruction. While the ND and MD domains could be modeled with high confidence, the poor density describing the endoU region of the apo-state reconstructions prevented accurate placement of the endoU domains. To determine whether there were any differences in the ND and MD domains, we calculated the RMSD between the UTP-bound wt-Nsp15 model and the models derived from the apo-state maps. Across all the models from this work, the RMSD of the ND and MD domains were in the range of 0.2-0.3 Å (see below). We added a statement to the results section to clarify this point.

Reconstruction 1	Reconstruction 2	Nsp15 Domain	RMSD
wt-Nsp15	UTP-bound wt-Nsp15	ND	0.21 Å
Nsp15 H235A, dataset i	UTP-bound wt-Nsp15	ND	0.33 Å
Nsp15 H235A, dataset ii	UTP-bound wt-Nsp15	ND	0.23 Å
wt-Nsp15	UTP-bound wt-Nsp15	MD	0.21 Å
Nsp15 H235A, dataset i	UTP-bound wt-Nsp15	MD	0.32 Å
Nsp15 H235A, dataset ii	UTP-bound wt-Nsp15	MD	0.22 Å

1.2 Furthermore, whether the structure of the complex of Nsp15 and 5'-UMP has been resolved, if so, please show the electron density of 5'-UMP.

We apologize if it was unclear that we could visualize the electron density for the 5'UMP. In Supplementary Movie 1 we show the full cryo-EM reconstruction and zoom into the active site with the 5'UMP. This movie illustrates the electron density surrounding the 5'-UMP as well as important active site residues. There is also a panel in Supplementary Fig. 3f which shows the density for UMP.

1.3 If the conformational change of SARS-CoV-2 nsp15 is related to its function, is there an increase in the proportion of particles with significant conformational changes in Fig S6 where the nucleic acid substrate exists?

Each Nsp15 dataset converged onto one discrete conformational state, therefore we did not see a distribution of particles in different conformational states. However, the three apo datasets have conformational heterogeneity within the endoU domain suggesting that it samples a continuous conformational spectrum. We have clarified this point in the results.

1.4 In Fig 6, please show the structure of 5'-UMP-bound protomer and its 5'-UMP-free form from hexamer systems, and compare the structures. At the same time, please mark the RNA binding channel in the nsp15 hexamer structure.

We have added an additional panel to Supplementary Figure 4 with a small di-nucleotide modeled into the Nsp15 active site. Aside from the active site there are no large electropositive patches on the surface of Nsp15 making it difficult to predict an RNA binding channel beyond the electropositive active site pocket. Rather than show the MD derived models in Figure 6, we included an additional analysis of the RMSDs of critical active site residues from all the MD derived models. This information can be found in Supplementary Table 2.

1.5 Finally, the core of this research is the discovery of dynamic changes in the SARS-CoV-2 nsp15 hexamer (Fig S6-S8). However, it is not clear dynamic changes of these maps from which particle conformations. Please show the electron density of these different particle conformations with conformation changes in Fig S6-S8. In addition, please explain the correlation between these conformational changes and Nsp15' function, and how nsp15 can perform the function of endoribonuclease through these conformational changes. What factors drive these changes?

We did not observe any discrete states representing well populated conformational states of the endoU domain in the apo datasets. The 3D variability analysis suggests that the endoU domain samples a continuum of conformational states in the absence of ligand. The overlay of the first and last maps calculated from 3D variability are shown in Figure 5A.

The reviewer raises an excellent question about how can Nsp15 perform its catalytic activity through these conformational changes and what drives these changes. The short answer is that we do not yet understand the significance of these conformational dynamics. In the discussion we speculate on a few possible reasons for this flexibility such as allosteric communication between the 6 active sites and regulation of nuclease activity.

Other comments:

1.6. Fig1 is just a summary of early literature reports, it should be added to supplementary materials (lines 52, 75)

Following the reviewer's suggestion, we have moved Figure 1 to the supplementary materials.

1.7. Following 3D classification, the particles converged into a single prominent class with a resolution of 3.36 Å. However, the overall resolution is 3.38 Å (Fig. S2D), 3.72 Å (Fig. S2G) and

3.36 Å (Fig. S2G). Please explain what the difference between these different resolutions means. (lines 106-107); Similar to Fig S6

We thank the reviewer for catching this mistake. We have updated Fig S2 (now Supplementary Fig. 3) and S6 (now Supplementary Fig. 7) with the correct resolution values.

1.8. *Coronavirus Nsp15 is a protein with conserved structure and function. It is recommended to compare SARS-CoV-2 Nsp15 with the resolved structure of coronaviruses nsp15, and provide RMSD value (line 119). Besides, SARS-Cov-2 is SARS-CoV-2*

Thank you for this suggestion, we have calculated the RMSDs between SARS-CoV-2 Nsp15 and other Nsp15 structures from the PDB. We do not report these values in the manuscript as a similar analysis was recently published with the crystal structure of SARS-CoV-2 Nsp15 (PDB ID 6VWW; Kim et al Protein Science 2020) and our values are in good agreement with their analysis.

Nsp15 source	PDB code	RMSD of chain A vs. UTP-wt-Nsp15 (Å/# residues aligned)
UMP-bound SARS-CoV-2	6WLC	0.36/345
Apo-state SARS-CoV-2	6VWW	0.37/345
SARS-CoV-1	2H85	0.51/345
MERS	5YVD	1.16/337
WT MHV	2GTH	1.20/329
HCoV-229E	4RS4	1.57/329
XendoU	2C1W	2.79/132

1.9. *Lines 127: It seems that 5'-UMP has not observed the corresponding electron density through cryo-electron microscopy, but the analysis of the effect of 5'-UMP and SARS-CoV-2 nsp15 active site is completed through molecular docking. Furthermore, the analysis of the active sites of 5'-UMP and Nsp15 did not clarify new ideas compared with previous literature reports. Besides, please compare the difference between SARS-CoV nsp15 and SARS-CoV-2 nsp15 nuclease activity*

We apologize if it was unclear that we determined the structure of Nsp15 bound to 5'-UMP and also carried out molecular docking. Supplementary Movie 1 and Supplementary Figure 3f show the density in the active site. We agree that the Nsp15 active site reported here is in excellent agreement with previous literature reports. However, our manuscript reveals new information about Nsp15 specificity, cleavage products, and dynamics of the nuclease domain. Most strikingly, our cryo-EM datasets and molecular dynamics simulations revealed the dynamics of the endoU nuclease domain.

Aside from a preference for cleaving 3' of uridines we don't yet know much about the nuclease activity of Nsp15 from SARS-CoV-2. Based on the high sequence and structural similarity with SARS-CoV-1 we hypothesized that the nuclease activity will be very similar, but no direct comparisons have been made.

1.10. *Line226 Fig. 3F is Fig. 4F*

Compared with the structure of RNase A active sites, it is found that H235 from Nsp15 and H119 from RNase A do not superimpose, are the structures of other coronavirus Nsp15 similar?

Please further explain how the weakening of nsp15 activity affects the invasion and proliferation of the virus itself.

We thank the reviewer for this suggestion and have included additional panels comparing the SARS-CoV-2 Nsp15 active site with other metal-independent endoribonucleases such as RNase A, Angiogenin, and other Nsp15 homologs (please see Fig. 4).

Thanks to the suggestion of the reviewer we also cite additional studies in the introduction which link Nsp15 nuclease activity to proliferation of the virus. While Nsp15 is not required for viral RNA synthesis, loss of Nsp15 nuclease activity has been shown to provide a protective immune response in both mice and pigs. For example, work from the Baker Lab has established that mouse hepatitis virus (MHV) with an endoU deficient Nps15 show a significant attenuation in mice compared to the wild type virus. This attenuation is the result of the activation of host immune responses (for a comprehensive review on Nsp15 please see Den and Baker, *Virology*, 2018).

1.11. Why are there differences in the Protein expression between Wt-Nsp15 and Catalytic-deficient Nsp15 variants (H235A or H250A)? (lines 351-361)

This is an excellent question. Wt-Nsp15 appears to be toxic to *E. coli*. It took extensive optimization to be able to purify sufficient quantities of Wt-Nsp15. In contrast the catalytic deficient mutants of Nsp15 are not toxic and express at a much higher rate. This same phenomenon was observed with SARS-CoV-1 Nsp15 (Guarino *et al* *JMB* 2005).

1.12. Through cryo-electron microscopic analysis, the author found that there is a swing between the SARS-CoV-2 nsp15 monomer molecules, and believes that this swing may be involved in the binding of RNA. Generally, RNA is negatively charged. Is the RNA-binding region between nsp15 monomers identified in the article positively charged? Furthermore, the observed mutual movement between the monomers is very slight. Is there room for single-stranded RNA? In addition, nsp15 participates in the formation of the replication complex in the process of coronavirus replication. Does the movement between its monomers facilitate the interaction with other replication-related proteins? At present, through data analysis, is it possible to obtain the structure of different conformational states of nsp15 swinging between monomers? It is not clear that how the large swing is observed in the movies. If the resolution is very low, by continuously displaying maps of different conformations, does it truly represent the existing conformation changes?

Aside from the di-nucleotide modeled into the active site it is difficult to predict a large RNA binding interface within Nsp15 because Nsp15 on the whole is very negatively charged (please see Supplementary Fig 4c). Work with MERS has shown that Nsp15 associates with Nsp7 and Nsp8, which are both part of the replication complex. Previous work revealed that Nsp7 and Nsp8 enhance Nsp15 activity (Zhang *et al* *J Virol*, 2018). Nsp8 has a large RNA binding surface so we hypothesize that Nsp8 may facilitate RNA recruitment to Nsp15 but this awaits experimental validation. Given that we do not know any details about the putative interface between Nsp15 and Nsp7/Nsp8 its difficult to speculate how endoU movement may influence their association. The 3D variability analysis does not reveal discrete conformational states. This analysis suggests that the endoU domain samples a conformational continuum when not bound by a ligand. We agree with the reviewer that the motion is not dramatic which is why we refer to the motion as a wobble.

1.13. Whether coronavirus nsp15 can non-specifically cut the viral genome RNA and host RNA, if so, the loss of nsp15 activity does not seem to improve the replication ability of the virus, please discuss.

As mentioned above we have further elaborated on Nsp15's role in viral replication in the introduction. Nsp15 is not required for viral RNA synthesis but it is important viral propagation.

1.14. After the loss of coronavirus Nsp15 activity, the virus still seems to be able to replicate. Therefore, nsp15 may be a target for vaccine development, not a target for drugs.

The reviewer is correct that it has been shown that the different strains of the coronavirus can replicate without Nsp15 in cell culture, however loss of Nsp15 nuclease activity in animal models leads to attenuation of disease severity and activation of a protective immune response. Nsp15 could therefore be a target for vaccine development (weakened/attenuated virus) or a small molecule inhibitor. This is now mentioned in the introduction.

Reviewer #2 (Remarks to the Author):

In this manuscript Pillon, Frazier and Dillard et al. determine cryoEM single particle structures of the SARS-CoV-2 endoribonuclease Nsp15, study its enzymatic mechanism by substrate cleavage assays and provide insights into its dynamics and substrate binding properties. The authors show that Nsp15 in solution is organized into a hexamer, a dimer of trimers. The structure bound to UMP reveals details of the interaction with the substrate which allows the authors to speculate on the enzymatic mechanism. They utilize substrate cleavage assays followed by mass spectrometry to interrogate the enzymatic reaction, revealing the accumulation of 2'3'-cyclic phosphate cleavage products. The catalytic EndoU domain is more flexible in the absence of a substrate in cryoEM data-based structural variability analysis and MD simulations. This most likely helps efficient substrate interaction. The results presented here provide crucial novel insights into the mechanism of this key viral endonuclease, paving the way for potential future drug development and a deeper understanding of the molecular basis for SARS-CoV-2's life cycle and host interactions.

The study is well designed, the included data are of very high quality and the conclusions reached are supported by the data provided. The manuscript is very well written and the figures are informative and well structured. Before publication the authors should address a few points outlined in the following.

We thank the reviewer for their support of our manuscript.

Major points:

2.1) The authors should provide evidence for the correlation between their structural models and the respective 3D reconstructions as a measure of model confidence. This is typically done by a flavor of the CCmap vs model, the FSCmap vs model plot and cut off resolution at 0.5 and 0.143.

We thank the reviewer for this suggestion. We determined the CC map vs the model values using the Phenix software suite and have added this in Table 1. We have also added the resolution limit with an FSC cutoff of 0.5 in Table 1.

2.2) Do the authors see differences in the arrangements of the endoU active site of Nsp15 in the UMP-bound, apo and mutant 3D reconstructions they calculated?

This is a great question but unfortunately the poor quality of the density in the endoU domain prevented us from building/refining an accurate model for the endoU domain in the apo reconstructions. However, we did a rigid body dock of the endoU domain into each of the maps from the 3D variability analysis and the overlay of this docking is shown in Fig. 5. We also analyzed the changes in distances from active site residues based on our MD derived models (New Supplementary Table 2). This analysis suggests that the endoU domain does not undergo significant conformational rearrangements but stays fixed as a rigid body while it wobbles with respect to the ND and MD domains.

2.3) An additional figure panel illustrating the direct comparison between the Angiogenin active site with the active site of the UMP-bound state (and potentially again the RNase A active site) would be of great help for readers.

This is a great suggestion. We have created a new figure (Figure 4) where we compare the active site of Nsp15 with RNase A and angiogenin. We also compare the active sites of Nsp15 from other viruses.

2.4) How does the recent x-ray crystallographic structure(s) compare to the cryoEM structures determined here? Do the B-factors of the model based on the x-ray crystallographic data correlate with the domain movements/flexibility the authors observe in their variability analysis and MD analysis?

The RMSD between our UMP bound structure with the recently deposited UMP bound x-ray structure (PDB ID 6WLC) is ~ 0.35 Å (across all residues in chain A), thus the structures are very similar to one another. The B-factors of the UMP crystal structure do not correlate with the domain movements that we observed by our 3D variability and MD analysis; however, this supports our model and MD analysis which suggest that ligand binding locks the endoU domain in place. There is an unpublished deposited structure of Nsp15 from HCoV-299E (PDB ID 4RS4) that does not have a ligand, phosphate, or metal bound in the active site. In this structure the B-factors of the endoU domain are significantly higher than the rest of the protein (see below). This comparison of the B-factors is included in Supplementary Figure 11.

Ribbons were colored by average B-factor value with a blue threshold of 31.7, white threshold of 73.5, and red threshold of 115.3.

2.5) Do the authors expect UMP to function as an inhibitor of the cleavage reaction? Is there a straight forward way to test this? Data down these lines would further strengthen the manuscript significantly in light of future structure-guided drug development.

We thank the reviewer for this suggestion. We repeated our FRET-based nuclease assay in the presence of increasing concentrations of UTP and found that there is an inhibition effect but only at very high concentrations (mM range). Therefore, UTP can inhibit nuclease activity but not at a meaningful concentration for drug development, so we opted not to include this data in our main text. A recent pre-print revealed that tipiracil, which is a derivative of uracil, is also a modest inhibitor of Nsp15, suggesting that the uracil backbone may be a good starting point for the development of a more potent inhibitor.

2.6) How do the crucial active site residues the authors identify behave in their MD simulations? How do their distances to each other and the UMP moiety behave over the time course of the simulation?

We compared the distances between active site residues H235, H250, K290, V292, S294, Y343, and L346 from each MD simulation. This information is included in the revised manuscript in Supplementary Table 2. The values in the table are the distances averaged over three independent MD runs and the standard deviations are given in parenthesis. We do not observe any significant deviations for these distances in the presence or absence of ligand. This suggests that the endoU domain is a rigid body that loses its correlation with the rest of the Nsp15 molecule in the absence of ligand, resulting in an endoU 'wobble'. Our MD analysis also revealed interesting information about the UMP moiety. All 6 protomers in the hexamer initially bind to UMP but only one remains stably bound during the MD simulation (please see Supplementary Table 3).

Minor points:

A recent preprint published on bioRxiv (<https://doi.org/10.1101/2020.06.26.173872>) concludes

that Tipiracil inhibits SARS-CoV-2 Nsp15 by binding to the active site. How does the Tipiracil binding mode proposed in this preprint relate to the authors conclusions on the enzymatic mechanism of Nsp15?

Tipiracil is a uracil derivative and it binds into the Nsp15 active site in the same position as the uracil in our UMP bound structure. Tipiracil makes hydrogen bonding interactions with Ser294 and most likely functions as a competitive inhibitor. The position of tipiracil in the Nsp15 active site agrees with the suggested role of Ser294 for base selectivity. We now mention Tipiracil in the discussion.

Fig.3 C The authors might want to refrain from indicating potential hydrogen bonds by dotted lines in the figures and rather describe them as such in the text/figure legend. While these contacts are likely the overall estimated resolution of the reconstruction at 3.38 Å does not support strong graphic conclusions of this kind.

As suggested by the reviewer we have removed hydrogen bonds in Fig 3C (now Fig. 2c).

Fig. 2. B) The authors might want to include labels for the domain arrangement in at least one of the protomers of the hexamer again for ease of understanding the multimeric arrangement of the complex

We thank the reviewer for this suggestion and have added domain labels to Fig. S3 (now Supplementary Fig. 4)

Is the Arctica the authors used equipped with a post-column energy filter before the K2 camera (and did they use it)? If so the authors should include this information in the methods section

The Arctica used for data collection is not equipped with an energy filter was used. We now mention this in the methods.

Line 214 "...m/z/ 923.14..." should be replaced by "m/z 923.14"

This typo has been corrected.

Line 218 "... (Fig. 3E) ..." should be replaced by "(Fig. 4E)"

The figures have been renumbered and Fig. 3e is now the correct reference here, but thank you for catching this mistake.

Reviewer #3 (Remarks to the Author):

In this manuscript, the authors reported several cryo-EM structures of SARS-CoV-2 nsp15 in its apo form as well as in complex with UMP. In my opinion, the authors draw two major conclusions based on these structures. In the first one, they proposed the critical role of His235 and His250 in chemical catalysis of the endonuclease cleavage. Specifically, they suggest the mechanism, in which His250 serves as the general base to activate the attacking nucleophile in hydrolysis. Their subsequent biochemical assays on the H250A mutant validated the critical role of this residue. In the second conclusion, based on the analysis of several cryo-EM datasets and MD simulations, the authors showed that the binding of UMP can greatly reduce the conformational flexibility and lock the EndoU domain. Overall, this manuscript addressed an

interesting topic, and SARS-CoV-2 nsp15 could serve as a novel drug target for COVID-19. However, I am not fully convinced by their first conclusion, though their second conclusion is interesting. See below for my comments:

We thank the reviewer for their support of our manuscript and the suggestion to include replicates to ensure the significance of our MD analysis.

3.1. In this work, the authors solve the structure of nsp15 in complex with UMP, rather than with a RNA chain. Similar structures have already been reported in Ref. 11 solved via X-ray crystallography. The critical roles of residues His235 and His250 have also been identified in Ref 11. This reduces the novelty of the current work.

The reviewer is correct that other structures of Nsp15 have been reported, in particular Ref 11 reports the first crystal structure of Nsp15 from SARS-CoV-2. However, the published Nsp15 structures do not contain RNA or nucleotides bound in the active site. Earlier work with Nsp15 homologs from other coronaviruses has revealed the importance of the two active site histidines but we felt it was important to validate the role of these histidines in SARS-CoV-2 Nsp15. Our manuscript also sheds light on uridine specificity, the composition of Nsp15 cleavage products, and the dynamics of the endoU domain. Cryo-EM allowed us to see these dynamics which are muted in crystal structures due to the constraints of crystal packing. We believe these dynamics are an important aspect of how Nsp15 functions on a molecular level.

3.2. The authors proposed a catalytic mechanism for the endonuclease RNA cleavage. In this mechanism, His150 serves as the general base to activate 2'OH as the attacking nucleophile. However, this is only based on a modelled structure by simply introducing a 3'-PO4 into their nsp15-UMP structure (see Fig. 3C). This is not convincing to me. In my opinion, the binding of RNA chain will impact the orientation of its nucleotide in the active site, which could lead to a different conformation from the UMP+3'PO4. Therefore, I would suggest that the authors should attempt to model a short RNA chain in complex with nsp15 to further validate their proposed mechanism.

During the revision of this manuscript the Structural Genomics Group from the University of Chicago deposited a new structure of SARS-CoV-2 Nsp15 with a Uridine-2',3'-Vanadate (PDBID 7K1L). This unpublished structure is an excellent mimic of the cyclic phosphate intermediate. We superimposed these coordinates with our UMP coordinates and observed good agreement with the modeled phosphate and the vanadate (please see below). We agree with the reviewer that an uncleaved RNA substrate may impact orientation in the active site, but this awaits structure determination. In order to trap this state one will have to alter the RNA substrate or the active site to prevent cleavage, either of which could also impact the conformation of the active site. We did attempt to model a short RNA chain, but we could only confidently model a di-nucleotide RNA substrate due to the absence of an extensive positively charge surface (Supplementary Figure 4).

3.3 The MD simulation part is interesting. However, a lot of details of their simulation set-up are missing (see below for some examples):

- The number of waters in their simulation box
- The type and size of the box
- The number of counter ions added
- How to deal with coulomb and vdw interactions?
- How to choose the protonated states of protein residues.
- Parameters for the MMGBSA to compute free energy.

As suggested by the review we have included all of the above details of the simulation set-up in the methods.

3.4 My understanding is that the authors only perform a single MD simulation for each system. The authors need to perform additional MD simulations to show that their reported results are statistically significant.

As suggested by the review we repeated all MD simulations in triplicate. The additional replicates are in good agreement with the original simulations and these results are included in Supplementary Fig. 10-11.

REVIEWERS' COMMENTS

Reviewer #1 (Remarks to the Author):

The revised manuscript has taken into account of the reviewers comments. I recommend to seek an editorial checkup on a few typos and grammatical expression throughout the text.

Reviewer #2 (Remarks to the Author):

The authors, Pillon, Frazier and Dillard et al., have strengthened the manuscript by adding new data and additional in-depth discussion to address the concerns and questions raised in the first round of review to satisfaction.

The manuscript has gained further in strength and now presents a timely and very valuable molecular and structural analysis of the SARS-CoV-2 endonuclease Nsp15.

I recommend publication without further delay.

Minor comments:

Figure 1 C: the black box in place to highlight the active site currently blanks out this area of the model at least in the pdf available to the reviewers. This should be urgently resolved before publication.

Page 11, line 260: "...this data..." should be replaced by "...these data..."

Page 12 line 292: "Supplementary Fig. 4" should most likely be replaced by "Supplementary Fig. 5"

Page 13, line 323: sentence starting with "Finally..." – it should be made clear that the comparisons here are being drawn between residues in the active site of one protomer (to clearly differentiate this statement from what is said in Page 13, line 317)

Page 18, line 430/431 (Material and Methods section): "total dose/dose rate" should be replaced by "exposure per sec/total exposure" or "fluence/flux"

Reviewer #3 (Remarks to the Author):

In the revised manuscript, the authors have appropriately addressed my comments and I would like to recommend its publication. In addition, I would like to suggest the authors to include the superimposition of their modeled UMP-phosphate coordinates and the recently published Vanadate (PDBID 7K1L) as a SI figure.

NCOMMS-20-35675 Rebuttal

Reviewer comments (*blue italics*) and our response (black)

Reviewer #1 (Remarks to the Author):

The revised manuscript has taken into account of the reviewers comments. I recommend to seek an editorial checkup on a few typos and grammatical expression throughout the text.

We thank the reviewer for their support of our revised manuscript. We have carefully edited the manuscript and corrected minor typos and errors in grammatical expression.

Reviewer #2 (Remarks to the Author):

The authors, Pillon, Frazier and Dillard et al., have strengthened the manuscript by adding new data and additional in-depth discussion to address the concerns and questions raised in the first round of review to satisfaction. The manuscript has gained further in strength and now presents a timely and very valuable molecular and structural analysis of the SARS-CoV-2 endonuclease Nsp15.

I recommend publication without further delay.

We thank the reviewer for their support of our revised manuscript.

Minor comments:

Figure 1 C: the black box in place to highlight the active site currently blanks out this area of the model at least in the pdf available to the reviewers. This should be urgently resolved before publication.

This appears to be an issue with the pdf available to reviewers as this is not a problem with the high resolution version of the figure.

Page 11, line 260: "...this data..." should be replaced by "...these data..."

This has been corrected.

Page 12 line 292: "Supplementary Fig. 4" should most likely be replaced by "Supplementary Fig. 5"

Thank you for catching this error. The reference to the figure has been updated to Supplementary Fig. 5.

Page 13, line 323: sentence starting with "Finally..." – it should be made clear that the comparisons here are being drawn between residues in the active site of one protomer (to clearly differentiation this statement from what is said in Page 13, line 317)

We have updated this sentence to make this clear.

Page 18, line 430/431 (Material and Methods section): "total dose/dose rate" should be replaced by "exposure per sec/total exposure" or "fluence/flux"

This has been corrected.

Reviewer #3 (Remarks to the Author):

In the revised manuscript, the authors have appropriately addressed my comments and I would like to recommend its publication. In addition, I would like to suggest the authors to include the superimposition of their modeled UMP-phosphate coordinates and the recently published Vanadate (PDBID 7K1L) as a SI figure.

Thank you for this suggestion. We have now added this figure as “Supplementary Figure 6”.